# Classification and Prediction of Typhoon Levels by Satellite Cloud Pictures through GC–LSTM Deep Learning Model

**DOI:** 10.3390/s20185132

**Published:** 2020-09-09

**Authors:** Jianyin Zhou, Jie Xiang, Sixun Huang

**Affiliations:** 1College of Meteorology and Oceanography, National University of Defense Technology, Nanjing 211101, China; 15165381462@163.com (J.Z.); huangsixun2021@163.com (S.H.); 2State Key Laboratory of Satellite Ocean Environment Dynamics, Second Institute of Oceanography, State Oceanic Administration, Hangzhou 361005, China

**Keywords:** deep learning, GC–LSTM model, typhoon, satellite image, prediction system

## Abstract

Typhoons are some of the most serious natural disasters, and the key to disaster prevention and mitigation is typhoon level classification. How to better use data of satellite cloud pictures to achieve accurate classification of typhoon levels has become one of classification the hot issues in current studies. A new framework of deep learning neural network, Graph Convolutional–Long Short-Term Memory Network (GC–LSTM), is proposed, which is based on the data of satellite cloud pictures of the Himawari-8 satellite in 2010–2019. The Graph Convolutional Network (GCN) is used to process the irregular spatial structure of satellite cloud pictures effectively, and the Long Short-Term Memory (LSTM) network is utilized to learn the characteristics of satellite cloud pictures over time. Moreover, to verify the effectiveness and accuracy of the model, the prediction effect and model stability are compared with other models. The results show that: the algorithm performance of this model is better than other prediction models; the prediction accuracy rate of typhoon level classification reaches 92.35%, and the prediction accuracy of typhoons and super typhoons reaches 95.12%. The model can accurately identify typhoon eye and spiral cloud belt, and the prediction results are always kept in the minimum range compared with the actual results, which proves that the GC–LSTM model has stronger stability. The model can accurately identify the levels of different typhoons according to the satellite cloud pictures. In summary, the results can provide a theoretical basis for the related research of typhoon level classification.

## 1. Introduction

Typhoons have the widest damage range among all the natural disasters, which is the most invading disaster of coastal areas. In particular, storms caused by typhoons have caused huge losses to coastal ships and the marine industry [1]. According to statistics, economic losses caused by typhoons account for 1%–3% of Gross Domestic Product (GDP) [2]. The prediction of typhoons has always been a scientific issue in this field [3]. During the typhoon outbreak, it is difficult to obtain typhoon data directly from conventional climate and ocean monitoring data, which makes it difficult to predict typhoons [4]. With the improvement of the satellite remote sensing technology, meteorological satellite cloud pictures can more accurately and stably monitor the weather changes in real-time in all weathers, becoming the main means of observing and predicting typhoons [5]. Research on using satellite cloud pictures has achieved a series of results in the process of typhoon generation and development [6,7]. The influence of typhoons on people’s lives and property is closely related to its intensity. According to the satellite cloud picture analysis, different typhoon levels will cause different cloud clusters. Therefore, the levels of typhoons can be predicted according to the cloud cluster data transmitted by satellite cloud pictures [8]. The current research methods for typhoon level prediction are mainly divided into the subjective empirical method and the simulation analysis method [9]. The subjective empirical method requires professional knowledge; for example, the Dvorak analysis method is estimated by understanding the cloud system structure characteristics and specific parameters through empirical rules and constraints [10]. The simulation analysis method is the most used; atmospheric physical quantities such as different initial fields and boundary conditions are comprehensively considered to predict typhoon levels [11] The simulation analysis method depends on the accurate recognition of satellite cloud pictures, and the typhoon level prediction model has poor accuracy and large errors [12]. Therefore, the establishment of a high-precision typhoon level classification model is crucial for the study of typhoons.

As a classification and recognition method with strong generalization ability, deep learning overcomes the shortcomings of traditional methods that require prior knowledge to explicitly extract features. Of the latest years, many scholars worldwide have applied deep learning techniques to marine meteorological research. Krapivin et al. employed sequential analysis and seepage theory tools to analyze the process of the ocean–atmosphere coupling system; additionally, they adopted the SVM to detect the state features of this system; such a method helped to monitor the changes and directions of the ocean transition process and could predict significant changes in the state of the ocean–atmosphere system [13]. Varotsos et al. proposed an information modeling tracker for tropical cyclones based on the clustering algorithm to assess the instability of the atmosphere–ocean system; the synthesized functional prediction structure could be a reliable global ocean monitoring system, which could effectively reduce the risk of tropical cyclones [14]. Zhu et al. (2019) established a short-term heavy rain recognition model based on physical parameters and deep learning; this model could automatically predict the probability of heavy rain occurrence based on data from various monitoring stations [15]. Scher employed a deep neural network based on the principles of cyclic model dynamics; after training the model, the network could predict ocean weather several hours in advance [16]. Kaba et al. input astronomical factors, extraterrestrial radiation and climate variables, sunshine duration, cloud cover, minimum temperature, and maximum temperature as attributes to obtain a climate prediction model; the prediction accuracy of the model for the marine climate was 98% [17]. Xiao et al. used convolutional Long Short-Term Memory (LSTM) networks as building blocks and trained the blocks in an end-to-end manner to achieve accurate and comprehensive predictions of sea surface temperature in the short and medium term [18]. However, there is little research on the characteristics of using satellite cloud pictures to identify typhoon levels through deep learning technology [19]. The main reason is that traditional machine learning algorithms need to explicitly extract features for classification, but it is difficult to extract features related to typhoon level classification in satellite cloud pictures.

Therefore, the poor accuracy, complex satellite cloud picture feature extraction, and low recognition rate in typhoon prediction are the problems to be focused on here. By introducing a Graph Convolutional Network (GCN) and a Long Short-Term Memory (LSTM) neural network, a typhoon Graph Convolutional–LSTM (GC–LSTM) neural network is constructed. The GC–LSTM model uses satellite cloud picture data of 20 years as samples for deep learning, and is compared with traditional typhoon prediction models. The results are expected to provide accurate and fast weather information for relevant departments to help them make decisions and reduce human life and property losses caused by typhoons.

## 2. Materials and Methods

### 2.1. Satellite Cloud Images and Data Sources

The meteorological satellite cloud picture is to use the meteorological satellite instruments in space to photograph the earth’s atmosphere, to find the weather through some rules, and to verify the weather in combination with the ground weather [20]. It can display various types of clouds on a single picture, characterize weather phenomena at different scales, and provide very useful telemetry data for weather analysis and forecasting. Generally, satellite cloud pictures can be divided into infrared satellite cloud pictures, visible light satellite cloud pictures, and cloud pictures processed and synthesized according to requirements. Figure 1 below is a processed RGB color cloud picture.

Typhoons are the most common severe weather system. Violent winds above level 12, huge waves above 6–9 m, storm surges above 2–3 m, and heavy rains above 200–300 mm always accompany typhoons, which are harmful to marine ships, the marine engineering industry even people’s lives and properties in coastal and inland areas. The embryo of a typhoon, that is, the initial cyclonic low-pressure circulation, has the following sources: (1) low-pressure disturbance in the tropical convergence zone, accounting for about 80–85%; (2) easterly wind belt disturbance—easterly wind wave, accounting for about 5–10%; (3) westerly wind belt disturbance degeneration, accounting for about 5%; (4) low-level vortices induced by high-level cold vortices, accounting for less than 5%. There is a very favorable weather situation in the Northwest Pacific: a strong tropical easterly wind belt, a strong and active equatorial westerly wind belt, an active tropical convergence belt, and a southwest-southeast monsoon convergence belt. Hence, typhoons are particularly prone to generate. Before a typhoon is formed, it undergoes an enhanced development process, which usually develops from an existing tropical cloud cluster of 3–4 d. Tropical cloud clusters and isolated cloud clusters in the zonal cloud belt if they can maintain existence for more than 3–4 d and can present a cyclonic low-pressure circulation. Once the surrounding long cloud belts can form one or several convections and be involved in the low pressure, after l–2 d, the low pressure can develop into a tropical cyclone typhoon. Therefore, when identifying the satellite cloud picture, whether it meets the rules based on the cloud cluster features should be determined first. Then, whether it is a meteorological feature of a typhoon should be judged based on the overall cloud cluster. In this way, the satellite cloud pictures can effectively identify typhoons. However, traditional methods also depend on recognizing these features; without enhancement, features in the satellite cloud pictures cannot be compared accurately, thereby reducing the prediction accuracy.

The data come from the National Institute of Informatics (NII) of Japan. (1) The website is: http://agora.ex.nii.ac.jp/digital-typhoon/. The Japanese “Himawari” series of satellite cloud pictures are utilized. From 2010 to 2019, Japan successively launched the “Himawari” satellites. In particular, there are 16 “Himawari-8” geosynchronous weather satellites. For the band channel, the spatial resolution can reach up to 500 m; (2) Analysis area: The coverage range of satellite cloud picture data is the upper part of the Northwest Pacific (120° E–160° W); (3) Data time: The high-resolution satellite cloud picture data of this area have been downloaded, as well as the information of typhoon intensity; all the data were transmitted from the Himawari-8 satellite from 2010 to 2019 in Japan.

The model established here is based on Ubunt16.10. The processor is the computing node of the Beijing Supercomputing Center server. The hardware configuration is 2 channels and 32 cores, EPYC 7452 @2.35 GHz, and 256 GB memory. The deep learning framework used is open-source Keras. The dataset contains more than 1000 typhoon processes. The experiment uses infrared cloud pictures as data samples. The objective is to obtain all-weather meteorological data. According to the typhoon level index, different typhoon level labels are formulated, as shown in Table 1:

Cloud pictures processing: First, the median filtering is performed on the original infrared image to remove the noise in the cloud image, which effectively retains the edge information of the picture. Second, the nearest neighbor scaling is used to convert the cloud picture into a 24 × 24 × 1 format as input information. Then, according to the criteria in Table 1, the classification is performed, where A represents tropical depression, B represents typhoon, C represents strong typhoon, and D represents super typhoon. Finally, a dataset of 3500 training samples and 600 test samples is constructed. There are, respectively, 1000 training sets and 200 test sets for tropical depression, typhoons, strong typhoons, and super typhoons. Some satellite cloud picture samples are shown in Figure 2.

### 2.2. Traditional Convolutional Neural Network

Deep learning Convolutional Neural Network (CNN) is a feed-forward neural network, which can quickly respond to nearby covered networks through artificial neurons, thus achieving a deep learning algorithm for rapid response to data [21]. It consists of a convolutional layer and a sampling layer alternately forming a network topology [22]. CNN uses the method of backpropagating neurons to realize the update network of each neuron information. The feature extraction process can be expressed as a score function $S(x_{i},w,b)$. The cross-entropy loss function for the classification error of the *i* sample $\left(x_{i},y_{i}\right)$ is defined as:(1)$$L_{i}=–\ln e^{s_{y_{i}}}+\ln\sum e^{s_{i}}$$

In (1), *S_yi_* represents the number of scores for the true classification of the *i*-th sample of the training set, and *S_i_* indicates the ratio of the index of the current element to the sum of all the element indices. The output of sample $\left(x_{i},y_{i}\right)$ after passing through the network is f(x), and the corresponding sample loss value is:(2)$$L_{i}(f(x),y)=–\ln f{\left(x\right)}_{y}$$

The error-sensitive items of the output layer of the CNN of the deep learning layer l are:(3)$$\delta_{l}=\frac{\partial L}{\partial a^{l}}=\nabla_{a}^{l}(x)-\ln f{\left(x\right)}_{y}=f(x)-y$$
where $a^{l}$ represents the input of layer l. Finally, the backpropagation rule of CNN is used to update the weight of each neuron, which makes the overall error function of the model continuously decrease.

Figure 3 shows the convolution layer of CNN, which uses its convolution kernel to convolve with the input image and then outputs the feature image of this layer through the neuron activation function, which realizes the feature extraction of the image. The convolution process is defined as follows:(4)$$x_{j}^{l}=f(\sum_{i\in M_{j}}x_{i}^{l-1}\times k_{ij}^{l}+b_{j}^{l})$$
where l is the number of convolutional layers in the model, $k_{ij}^{l}$ is the number of convolution kernels, $b_{j}^{l}$ is the additive bias, f is the activation function, and $M_{j}$ is the input image. The specific structure process is shown in Figure 3:

Figure 4 shows the data collection layer of the convolution nerve. The data collection layer is the process of reducing the resolution of the current input feature image and reducing the amount of calculation, thereby improving the network convergence speed. The data collection can be defined as:(5)$$x_{j}^{l}=f(\beta_{j}^{l}down(x_{i}^{l-1})+b_{j}^{l})$$
where down(•) represents the data collection function, $\beta_{j}^{l}$ and $b_{j}^{l}$ represent the product bias and additive bias, respectively, f is the activation function. Among them, the characteristic of the sampling layer $C_{x}$ is 2 × 2 sampling, and every 4 pixels are combined into 1 pixel. Weighted by the multiplicative bias $w_{x+1}$, the addictive bias $b_{x+1}$ is output through the activation function $S_{x+1}$.

### 2.3. Deep Learning GCN Algorithm

The network topology of traditional CNN is formed by alternately arranged convolution layers and sampling layers. If the input features are not prominent, the pooling layer will lose some image information while reducing the dimensionality, reducing the network learning capability. The traditional CNN directly utilizes and compares the pictures with the original typhoon pictures. If the resolution of the transmitted data CNN image is low, the CNN prediction accuracy of the typhoon levels will decrease more. In the constructed dataset of satellite cloud pictures, due to the sophisticated atmospheric factors during typhoon formation, the spiral radius of the cloud pictures is not apparent. Therefore, traditional CNN is not suitable for feature extraction of typhoon cloud images. In contrast, the GCN network is a deep learning algorithm specially established based on images. It can extract the original image of the satellite cloud pictures according to some rules, effectively extract the features of the image, and improve the local image resolution; then, the model compares the processed image with the trained image database so that the processed image prediction accuracy is higher.

GCN is an algorithm that adds a lot of image processing based on CNN [23]. Most real-world network data is represented in the form of graphs, such as social networks, protein interaction networks, and knowledge graphs; however, they do not have a regular spatial structure and can process image data that cannot be processed by CNN. The algorithm transfers the convolution method on the image to the graph, and proposes two methods based on space and spectrum decomposition. The space-based method is to establish a perceptual domain for each node (selecting the neighbor node of the node); in other words, the nodes in the graph are connected in the spatial domain to achieve a hierarchical structure, so as to perform convolution learning. Based on the spectral decomposition method, the Laplacian matrix is used to transform the feature vector into the spectral domain; then, the point in the spectral domain is multiplied and the inverse Fourier transform is performed to achieve the convolution on the graph. The specific structure is shown in Figure 5. GCN is a CNN that directly acts on graphs. GCN allows end-to-end learning of structured data, and extracts the features of network nodes by learning the structural features of the network. Here, the GCN is utilized to extract the network structure features at each moment.

In the recognition of the typhoon cloud pictures, the general model directly utilizes the pictures and compares them with the original typhoon pictures. If the resolution of the transmitted data image is low, the prediction accuracy of the typhoon levels will decrease more. The GCN network is a deep learning algorithm specially established based on images. It can extract the original image of the satellite cloud pictures according to some rules, effectively extract the features of the image, and improve the local image resolution; then, the model compares the processed image with the trained image database so that the processed image prediction accuracy is higher.

The identification of satellite cloud pictures via GCN is as follows: (1) connecting a single sample of satellite cloud pictures to form a row vector; and (2) superimposing n vectors into the GCN system. Information can be transferred between nodes based on a correlation coefficient matrix. Therefore, the data-driven method constructs a correlation coefficient matrix, which contains the original satellite cloud picture’s image and edge features. This model uses the data-driven method to establish a directed graph between markers, and GCN maps the category markers to the corresponding category classifier. Finally, the category relationship is modeled, and at the same time, the model learning ability is improved. A correlation coefficient matrix is constructed for GCN by balancing the node and its neighboring nodes for node feature updates, thereby effectively solving the overfitting and excessive smoothing problems that hinder GCN’s performance.

### 2.4. LSTM Neural Network Algorithm

LSTM is a type of Recurrent Neural Network (RNN) network. It is often used to process and predict important events with very long intervals and delays in time series [24]. In typhoon prediction, accurate time prediction is very important. Therefore, the LSTM algorithm is chosen, which can accurately establish a time relationship graph. Based on the data transmitted from the traditional satellite and after feature extraction, the data will be arranged in chronological order. Second, according to the time interval of the previous typhoon, the algorithm can predict the time of the typhoon well. An LSTM unit contains an input gate, output gate, and forget gate. Among them, the input gate controls model input, the output gate controls model output, and the forget gate calculates the degree of forgetting of the memory module at the previous moment. The structure of the LSTM model is shown in Figure 6, the specific calculation is as follows:(6)$$f_{t}=\sigma (W_{f}\cdot \left[h_{t-1},x_{t}\right]+b_{f})$$
where $f_{t}$ and $i_{t}$ denote the forget gate and the input gate of the *t* step in the sentence sequence, respectively. In each sentence sequence, the forget gate controls the degree of forgetting the information of each word, and the input gate controls the degree to which each word information is newly written into long-term information.
(7)$$i_{t}=\sigma (W_{i}\cdot \left[h_{t-1},x_{t}\right]+b_{i})$$
(8)$$C=\tanh(W_{c}\cdot \left[h_{t-1},x_{t}\right]+b_{c})$$
(9)$$C_{t}=f_{t}\times C_{t-1}+i_{t}\times C$$

The two gates $f_{t}$ and $i_{t}$ use the Sigmoid function, the value range is [0, 1], and the value of tanh function is [−1, 1]. $C_{t-1}$ is the state of the neuron at time *t* − 1, and $C_{t}$ is the state of the neuron at time *t*.
(10)$$h_{t}=o_{t}\times \tanh(C_{t})$$
(11)$$o_{t}=\sigma (w_{o}\cdot \left[h_{t-1},x_{t}\right]+b_{o})$$
where $o_{t}$ is the output degree of the output-gate-controlled word long-term information $h_{t}$ is the output of step t in the sentence sequence. The above equations show that the word information of the current step of LSTM is determined by the word information retained in the previous step and the word information saved after being filtered by the input gate at the current time. Here, the LSTM network is introduced to effectively mine the long-term data and utilize the raw data information, to effectively process the cloud picture data.

### 2.5. Construction of the GCN–LSTM Fusion Model

The traditional CNN uses a topology structure composed of a convolutional layer and a data acquisition layer. When the input image features are not obvious, the resolution of the input picture will be actively reduced in the pooling layer of the network, which loses some important information of the pictures, resulting in a decline in network learning capabilities; in the meantime, the corresponding model accuracy will continue to decline.

The complex atmospheric factors in the formation of a typhoon make the features within the spiral radius of the cloud picture not obvious. In the data of satellite cloud pictures, every detail is very concerned. However, the traditional CNN is not very suitable for the feature extraction of typhoon cloud pictures. Based on the traditional CNN, the cyclic convolution is utilized to enhance the feature extraction capability of the model. By taking the advantages of the LSTM network, a novel deep learning model GC–LSTM is proposed to extract features in atmospheric cloud pictures to achieve accurate prediction of typhoons.

The entire GC–LSTM model combines the advantages of the two models: LSTM network and GCN. LSTM is used to learn the timing information of the connected state of each node, and GCN is used to learn snapshots at every moment. The structural characteristics of the network make it capable of effectively processing high-dimensional, time-dependent, and sparse structural sequence data. This model is easy to build and train, and can adapt to different network applications. Also, the accurate prediction of typhoon levels can be achieved. The specific structural framework is shown in Figure 7.

### 2.6. Model Verification and Optimization

(1)Model accuracy (*ACC*): It is the part that passes the true correct rate. If the number of real typhoons in the *i* sample of all n satellite cloud picture samples is y, and the data predicted by the model is *O_i_*, then the correct rate of the classification of the typhoon satellite cloud picture model is calculated as follows; if the number predicted by satellite cloud pictures is more consistent with the real number, the correct rate of model classification is greater.
(12)$$ACC(y_{i},O_{i})=\frac{1}{n}\sum_{i=0}^{n}1(y_{i}=O_{i})$$(2)Precision (*Pre*): It indicates the proportion of processed samples that are correctly divided into positive samples [25].
(13)$$Pre=\frac{N_{TP}}{N_{TP}+N_{FP}}$$
where *N_TP_* represents the number of satellite cloud pictures that should be correctly classified; *N_FP_* represents the number of true correct classification after passing through the typhoon satellite cloud picture prediction model.(3)Recall (*Rec*): It represents the proportion of positive samples in the original positive samples [26]. It indicates the proportion of the total number of correctly predicted numbers after the typhoon satellite cloud picture prediction model.(4)Recognition Rate (*RR*): It is the ratio of the wrongly recognized image/the recognized image [27].(5)Matching Speed (*MS*): It refers to the time from the completion of image acquisition to the completion of model prediction.
(14)$$Rec=\frac{TP}{TP+FN}\times 100\%$$
where *TP* is the number of typhoon searches that are correctly identified by satellite cloud pictures, and *FN* is the number of typhoon searches that are not correctly identified by satellite cloud pictures.

The model performance is compared. M1 stands for the ANN model, M2 stands for the RNN model, M3 stands for the GCN model, M4 stands for the LSTM model, M5 stands for the GCN–LSTM, and M6 stands for the RNN–LSTM model. RNN–LSTM has been employed in research [28]. Lian et al. (2020) utilized the RNN–LSTM model for predicting the path of satellite cloud pictures; the results revealed the high prediction accuracy of this model [29]. Zhao et al. (2020) proposed a typhoon identification and typhoon center location method based on deep learning; the average accuracy of this method was 96.83%, and the average detection time of each sample was 6 ms, which met the real-time typhoon eye detection. At present, typhoon paths are often predicted by the single algorithms or the CNN + SVM/LSTM algorithm. Although some of these algorithms have high accuracy, the output results are unstable, or the operation requires a higher configuration, which reduces the recognition speed [30]. No one has systematically summarized and compared the problems of the fusion algorithms, nor utilized the combination of GCN–LSTM for typhoon type prediction. The optimal model parameters mainly determine the convolution number of the CNN and the proportion of neurons. Among them, the convolution number of the input data is set to Q1 (1 × 1), Q2 (3 × 3), Q3 (5 × 5), Q4 (7 × 7), Q5 (8 × 8), and Q6 (9 × 9); the proportion of different neurons is set between 0% and 90%, and increases by 10% each time; the optimal parameter setting of the model is judged by its accuracy.

## 3. Results

### 3.1. Performance Analysis of Different Models

The performance of different models is shown in Figure 8. In terms of accuracy, the accuracy of the training set is higher than that of the test set, which is 5% higher on average. Comparing different models, it is found that as the number of training sets increases, the accuracy of the model continues to increase, which is consistent with the actual situation. When the training data are more, the accuracy of the model is higher. Compared with the single algorithm, the accuracy of the model is lower than the fusion model. The single model with the highest accuracy rate is the LSTM model. Because of its data memory function, the overall accuracy is significantly higher. Compared with other models, the average accuracy rate is 88.21%, and the fusion model with the highest accuracy rate is the GCN–LSTM model, with an average accuracy rate of 91.51%. In terms of accuracy and recall rate, the results are consistent with the conclusion of the accuracy rate. The highest is the GCN–LSTM model, with an average recall rate of 91.04% and an average accuracy of 92.35%. In terms of the recognition rate, the model shows a higher advantage, and the average recognition rate is as high as 93.61%. In terms of matching speed, the model has a stronger ability to recognize satellite cloud pictures than other models. Due to the advantages of the GCN and the rapid processing of data by LSTM, its average processing speed is maintained at 25.5 ms. According to the above results, the constructed GCN–LSTM model has a higher recognition rate for satellite cloud pictures, and the accuracy is 13.3% higher than that of the traditional ANN network. The classification effect of satellite cloud pictures is also significantly enhanced. The model shows a strong advantage in typhoon recognition.

### 3.2. Determination of Optimal Model Parameters

It is illustrated in Figure 9 that by using different convolutional numbers based on the original model, the convergence speed of the model differs greatly. In the training data set, when the convolution kernel number is 1 × 1, the loss value of the model is 0.55. When the convolution kernel number is 3 × 3, the model convergence speed is the slowest, and the stability can only be achieved when the number of iterations is 350 times. When the convolution kernel number is 7 × 7, the model converges and reaches stability at 310 times, and the loss value is 0.018. When the convolution kernel number is 5 × 5, the optimal loss value for the model convergence at 320 times is 0.0094. When the convolution kernel number is 8 × 8, the optimal loss value of the model at 300 times of convergence is 0.035, the overall convergence iteration number of the test set is larger, and the optimal loss value is also larger than the training set. In summary, 5 × 5 is chosen as the optimal convolution kernel number.

It is illustrated in Figure 10 that compared to the convolution kernel numbers of 1 × 1 and 3 × 3, when the convolution kernel number is 5 × 5, the neural network has a high processing efficiency for the satellite cloud picture. The 5 × 5 convolution kernel has obvious feature extraction effects for typhoon eye, cloud wall, and spiral cloud belt (yellow area), but the 8 × 8 convolution kernel will increase typhoon similar redundancy and thus lose local features. Compared with the original image of the vortex cloud area, it is found that the model is more sensitive to the yellow area, but not sensitive to the fibrous cloud at the edge of the typhoon. Therefore, adding the model to the yellow area feature extraction is beneficial to the classification of the model. This is consistent with the truth that many scholars predict typhoons by locating typhoon eyes, segmenting dense cloud areas, and extracting spiral cloud belt features. This also proves the feasibility of classifying typhoon levels through deep learning and satellite cloud pictures.

By using different convolution kernel numbers based on the original model, the accuracy of the model is not much different, which is shown in Figure 11. In the training data set, when the convolution kernel number is 1 × 1, the model accuracy rate is 83.8%; when the convolution kernel number is 3 × 3, the model accuracy rate is 87.2%; when the convolution kernel number is 7 × 7, the model accuracy rate is 93.3%; when the convolution kernel number is 5 × 5, the model accuracy rate is 97.1%; when the convolution kernel number is 8 × 8, the model accuracy rate is 91.9%; the overall accuracy of the test set is lower than the training set. In summary, 5 × 5 is chosen as the optimal number of convolutions for the model.

The accuracy of the model under different neuron ratios is shown in Figure 12. As the proportion of neurons continues to increase, the accuracy of the model shows a slight increase and then decreases. When the proportion of neurons is 30%, 50%, 80%, and 90%, the accuracy rate of the model drops; the lowest accuracy rate is 91% when the proportion of neurons is 30%, and the highest recognition rate is 92.5% when the proportion of neurons is 70%. The overall model accuracy of the test set is lower than that of the training set.

### 3.3. Application Analysis of Model Examples

It is shown in Table 2 that using the GC–LSTM model under the optimal parameters, the prediction accuracy rate of typhoons and super typhoons can reach 95.12%, but the prediction accuracy rate of tropical depressions is only 83.36% lower, which may be that the specific cloud characteristics and complete cloud structure are not formed during the typhoon formation process. Approximate results can also be seen in the results of some sample satellite cloud pictures. Therefore, the accuracy of the model is low. For strong typhoons, the accuracy rate is 93.24%, which is 2% higher than the traditional typhoon prediction model [31]. In summary, the GC–LSTM typhoon prediction model has high prediction performance, and the prediction accuracy of typhoons and super typhoons can reach 95.12%.

Figure 13A,B shows the data prediction of the GCN–LSTM model from 2010 to 2019. After the results are compared with actual values, the annual average absolute error is obtained. The result shows that after 6 h of model training, the average absolute error remains between 1 and 15; after 12 h of training, the average absolute error of the model drops by 33.33% and remains between 1 and 10, which further shows that the accuracy of the GCN–LSTM model is improved greatly.

Figure 13C illustrates that LSTM, the best single model, and RNN–LSTM and GCN–LSTM, two best fusion models, are employed for prediction. Act represents the actual output result on the test dataset, and Pre represents the actual output result on the training dataset. All satellite cloud pictures, from 2010 to 2019, are output and analyzed according to different years. The results show that when a fixed amount of data is input, the LSTM test set’s result is maintained between 30 and 253. Data fluctuates sharply in different years. Therefore, the model is unstable. Compared to the RNN–LSTM model, this model’s stability also has a large deviation due to the need for a large number of neural networks to perform operations. The prediction result interval of GCN–LSTM is 38–104, with the least data fluctuation. The above results show that the proposed GCN–LSTM model is more stable.

## 4. Conclusions

Through the analysis of the original typhoon prediction model, problems such as poor prediction accuracy, low recognition rate, and complex feature extraction are found in the traditional typhoon prediction models. Therefore, based on the deep learning neural network framework, the advantages of GCN and LSTM are utilized to build the typhoon GC–LSTM model to effectively process the problems in the construction of satellite cloud picture models. The proposed model is significantly better than other prediction models in terms of algorithm performance. Compared with the traditional ANN model, it has improved by 13.3%. The prediction accuracy of typhoons and super typhoons through optimization of parameters has reached 95.12%. It can accurately identify typhoon eyes and spiral cloud belts, and its stability is better. This model can provide a theoretical basis for typhoon prediction related research. Although the construction process and the actual application effect of the model have been elaborated on as much as possible, due to the objective limitations, the following deficiencies are found: (1) only one GCN image processing neural network is used, and no other image legend recognition algorithm is used for processing; (2) for the actual application of the model, only the prediction effect of the model is comprehensively evaluated, but actual application analysis with a large amount of data is not involved. In the future, in-depth research will be continued in these two areas, with a view to truly applying this satellite cloud picture-based typhoon prediction model to actual analysis, thereby reducing the impact of typhoons on people’s lives and property.

## Figures and Tables

**Figure 1 sensors-20-05132-f001:**
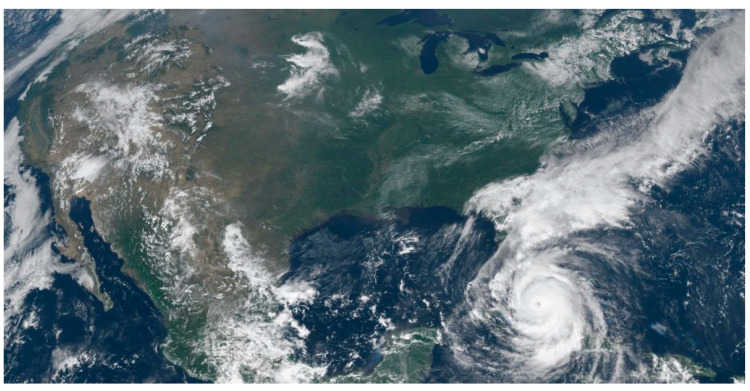
Meteorological satellite cloud picture.

**Figure 2 sensors-20-05132-f002:**
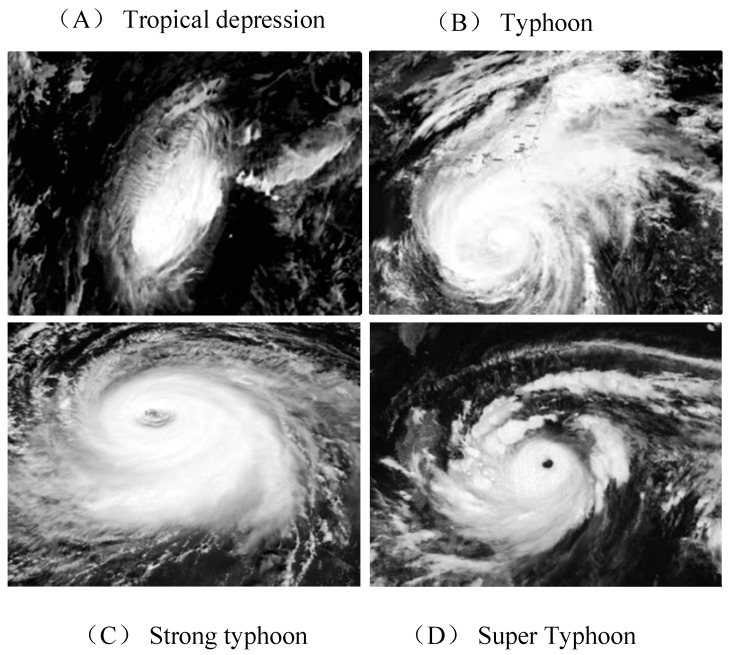
Some meteorological satellite cloud picture samples. Please add explanation of subfigures in caption.

**Figure 3 sensors-20-05132-f003:**
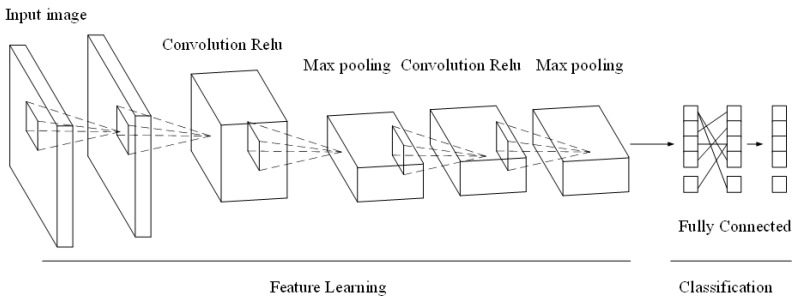
Structure of the deep learning convolution layer.

**Figure 4 sensors-20-05132-f004:**
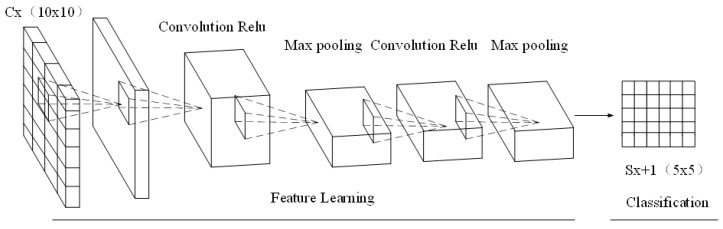
Structure of the deep learning collection layer.

**Figure 5 sensors-20-05132-f005:**
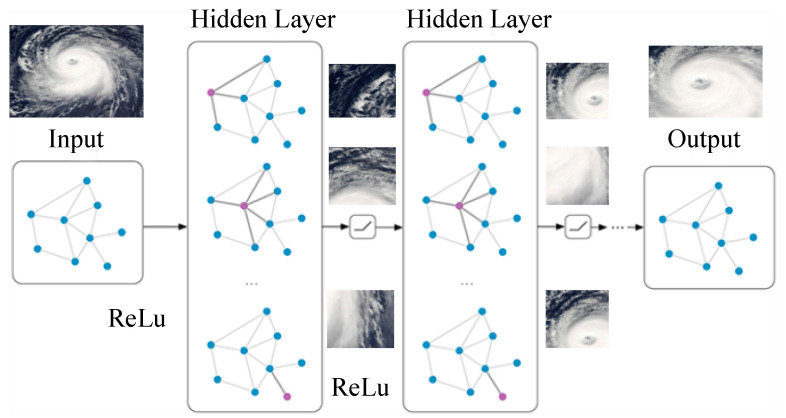
Schematic diagram of Graph Convolutional Network (GCN) structure.

**Figure 6 sensors-20-05132-f006:**
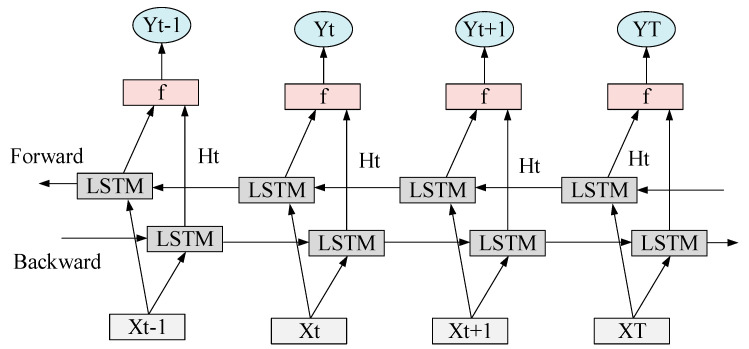
Schematic diagram of the Long Short-Term Memory (LSTM) model structure.

**Figure 7 sensors-20-05132-f007:**
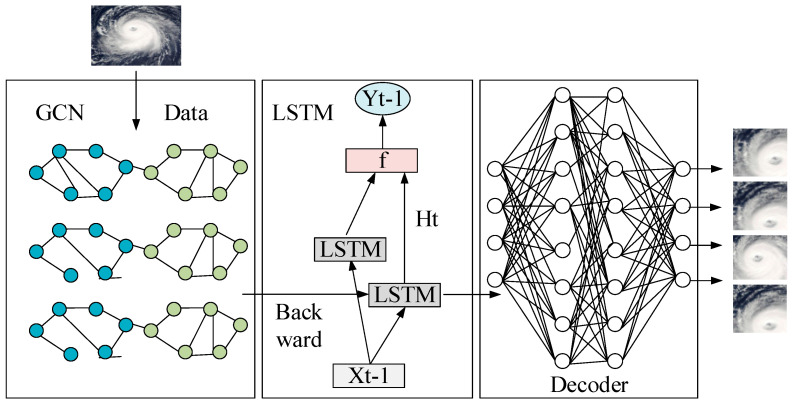
Overall framework of the GCN–LSTM fusion model.

**Figure 8 sensors-20-05132-f008:**
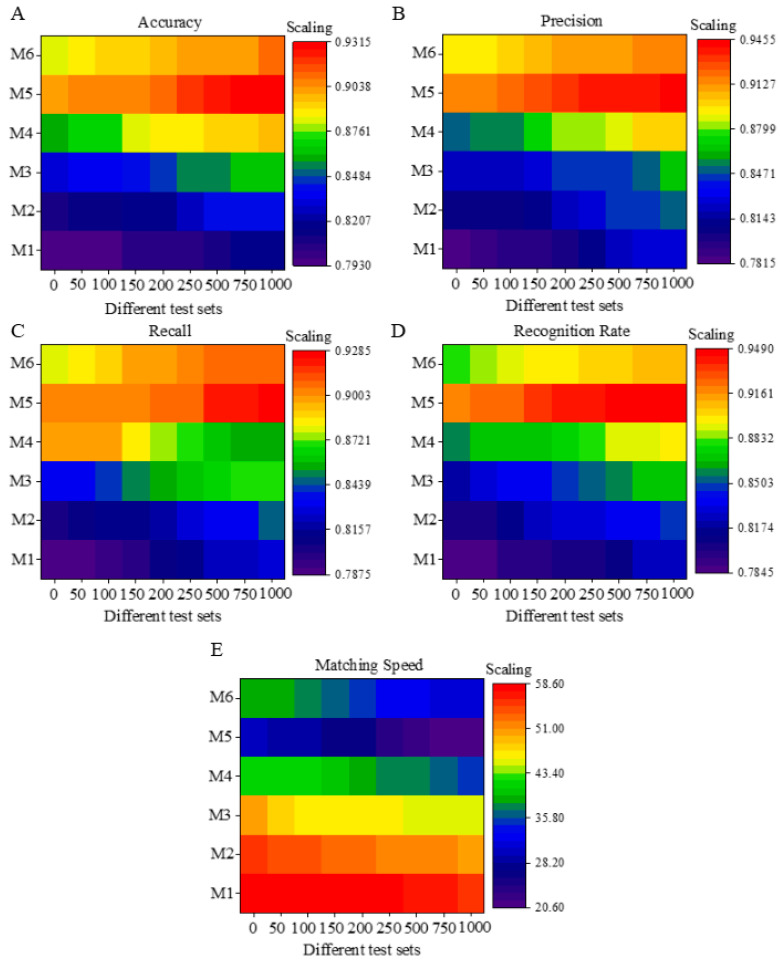
Performance comparison results of different models (Note: M1 is the ANN model, M2 is the Recurrent Neural Network (RNN) model, M3 is the GCN model, M4 is the LSTM model, M5 is the GCN–LSTM, and M6 is the RNN–LSTM model, where 0–200 is the test set and 200–1000 is the training set). Figures (**A**–**E**) show the performance results of different algorithms in accuracy, precision, recall, recognition rate, and processing speed under different test sets.

**Figure 9 sensors-20-05132-f009:**
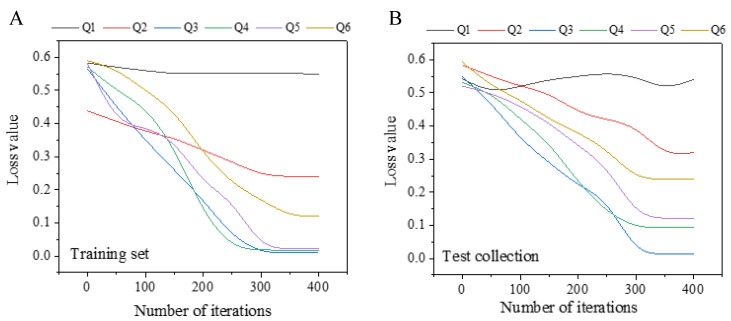
Feature extraction effect of different number of convolution kernels (Note: Q1 (1 × 1 convolution kernel), Q2 (3 × 3 convolution kernel), Q3 (5 × 5 convolution kernel), Q4 (7 × 7 convolution kernel), Q5 (8 × 8 convolution kernel), and Q6 (9 × 9 convolution kernel)). Figure (**A**) shows the feature extraction effect under different numbers of convolution kernels on the training set; Figure (**B**) shows the feature extraction effect under different numbers of convolution kernels on the test set.

**Figure 10 sensors-20-05132-f010:**
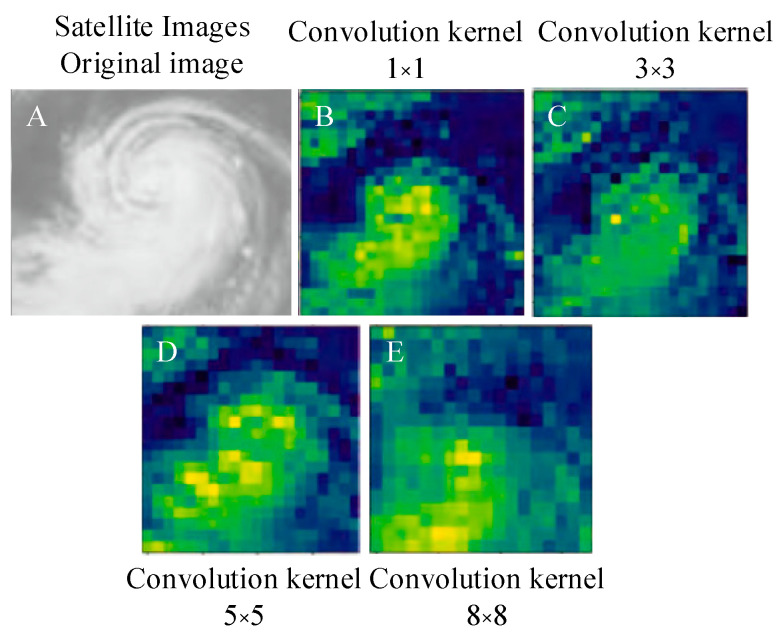
Feature extraction result of some convolution kernels.

**Figure 11 sensors-20-05132-f011:**
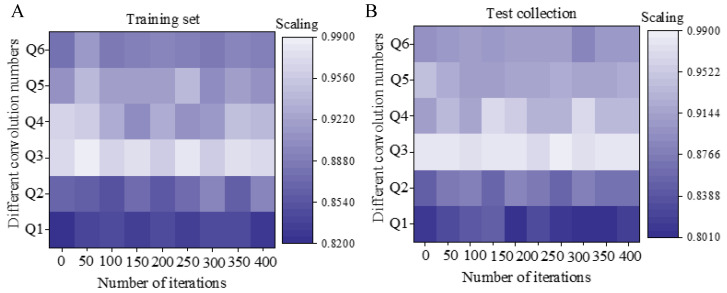
Model accuracy under different convolution kernels. Figure (**A**) shows the accuracy of the model under different convolution kernels on the training set; Figure (**B**) shows the accuracy of the model under different convolution kernels on the test set.

**Figure 12 sensors-20-05132-f012:**
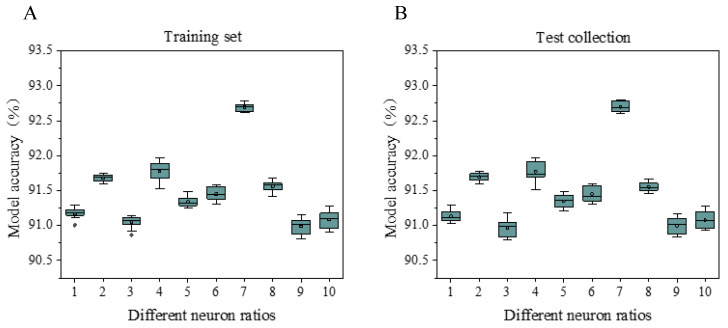
Model accuracy rate at different neuron ratios. Figure (**A**) shows the accuracy of the model under different neuron ratios on the training set; Figure (**B**) shows the accuracy of the model under different neuron ratios on the test set.

**Figure 13 sensors-20-05132-f013:**
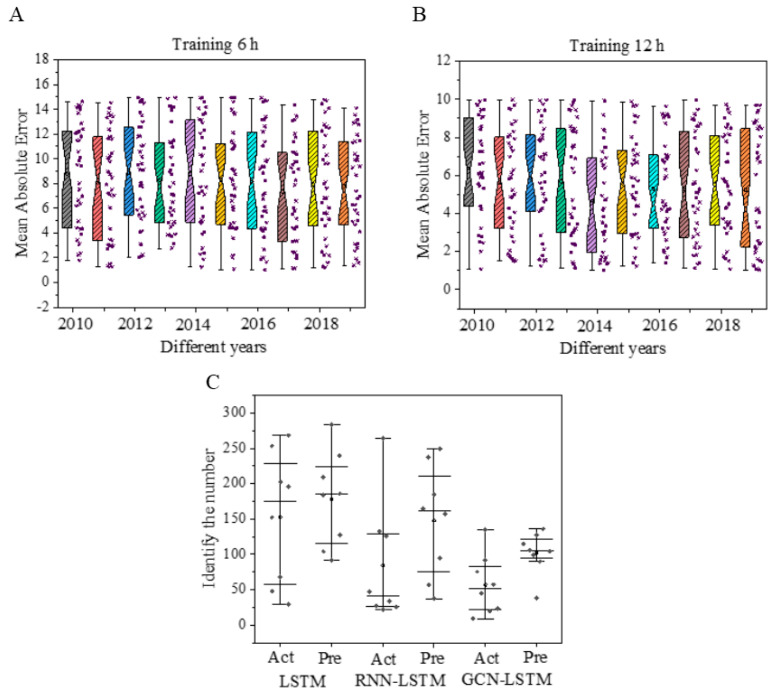
Model stability test results. Figure (**A**,**B**) illustrate the mean absolute errors of the GCN-LSTM model after 6 h and 12 h training on the data of 2010–2019, respectively. Figure (**C**) shows the identified and predicted number of typhoons of LSTM, RNN-LSTM, and GCN-LSTM models.

**Table 1 sensors-20-05132-t001:** Typhoon level standard label.

Typhoon Level	Maximum Wind Speed/kt	Maximum Wind Speed/(m/s)
Tropical depression	<34	<17
Typhoon	>34–<64	>17–<33
Strong typhoon	>64–<85	>33–<44
Super Typhoon	>85–<105	>44–<54

**Table 2 sensors-20-05132-t002:** Comprehensive evaluation of typhoon level prediction.

Categorical Data	Tropical Depression (0-)	Typhoon (1-)	Strong Typhoon (2-)	Super Typhoon (3-)
Tropical depression	83.36	12.67	9.59	3.28
Typhoon	1	95.12	0	0
Strong typhoon	1	1	93.24	7.24
Super Typhoon	0	0	1	95.12

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
