# Peer review of "Classification and Prediction of Typhoon Levels by Satellite Cloud Pictures through GC–LSTM Deep Learning Model"

_sensors, 2020, doi:10.3390/s20185132_

Round 1

Reviewer 1 Report

The paper proposed a GC-LSTM model to classify typhoon levels with satellite images as input. The contents were well written. There are some concerns with the technical aspects of the paper:

  1. There were some missing references, such as: 1) A hybrid CNN-LSTM model for typhoon formation forecasting.2019, 2) A Novel Data-driven Tropical Cyclone Track Prediction Model Based on CNN and GRU with Multi-dimensional Feature Selection.2020, 3) A Deep Learning Based Method for Typhoon Recognition and Typhoon Center Location.2019, 4) Machine Learning in Tropical Cyclone Forecast Modeling-A Review.2020. Particularly some of these paper also used convolutional and recurrent network to predict typhoon / tropical cyclone, which are highly relevant to the proposed method. Without formal comparisons to existing relevant methods, it is difficult to confirm the novelty of the proposed method.
  2. The use of graph convolutional network was unclear. The paper did not discussed and explained what were the unstructured data the proposal method tried to learn and represent. Why is a generic convolutional network not sufficient to user, particularly only satellite images (with regular structure) are used as the inputs?
  3. The experiment results did not compare the proposed method with existing relevant methods, but it only performed comparison with traditional ANN. Without such a comparison, it is not clear whether the proposed method really stands out from the existing similar approaches.

Author Response

Point 1: There were some missing references, such as: 1) A hybrid CNN-LSTM model for typhoon formation forecasting.2019, 2) A Novel Data-driven Tropical Cyclone Track Prediction Model Based on CNN and GRU with Multi-dimensional Feature Selection.2020, 3) A Deep Learning Based Method for Typhoon Recognition and Typhoon Center Location.2019, 4) Machine Learning in Tropical Cyclone Forecast Modeling-A Review.2020. Particularly some of these paper also used convolutional and recurrent network to predict typhoon / tropical cyclone, which are highly relevant to the proposed method. Without formal comparisons to existing relevant methods, it is difficult to confirm the novelty of the proposed method.

Response 1: Thanks for your comment. We have supplemented the missing references of relevant algorithms and deep learning to clarify the novelty of the algorithm we proposed. The details are presented in Section 2.6. At present, most of the researches focuses on the combination of CNN and other image recognition algorithms for typhoon path prediction. Nevertheless, research on the recognition of typhoon types is rare. Second, no one has systematically summarized and compared the problems of the fusion algorithms, nor utilized the combination of GCN+LSTM for typhoon type prediction. In this paper, we have compared our algorithm with other algorithms, and finally, proved that the performance of the GCN-LSTM fusion algorithm is optimal, which is the innovation of this paper.

Point 2: The use of graph convolutional network was unclear. The paper did not discussed and explained what were the unstructured data the proposal method tried to learn and represent. Why is a generic convolutional network not sufficient to user, particularly only satellite images (with regular structure) are used as the inputs?

Response 2: Thanks for your comment. We have already explained this issue in detail; please refer to Section 2.3 for details. First, the network topology of the traditional convolutional neural network is formed by alternately arranged convolution layers and sampling layers. If the input features are not prominent, the pooling layer will lose some image information while reducing the dimensionality, reducing the network learning capability. Second, in recognition of the typhoon cloud pictures, the traditional convolutional neural network directly utilizes and compares the pictures with the original typhoon pictures. If the resolution of the transmitted data image is low, the prediction accuracy of the typhoon levels will decrease more. In contrast, the graph convolutional network can extract the original image of the satellite cloud pictures according to some rules, effectively extract the features of the image, and improve the local image resolution; then, the model compares the processed image with the trained image database so that the processed image prediction accuracy is higher. Hence, we choose GCN for predictive analysis.

Point 3: The experiment results did not compare the proposed method with existing relevant methods, but it only performed comparison with traditional ANN. Without such a comparison, it is not clear whether the proposed method really stands out from the existing similar approaches.

Response 3: Thanks for your comment. First, we compare the existing GCN-LSTM with AR-LSTM, 3DCNN-LSTM, and RNN that are frequently discussed. These algorithms are the newest algorithms optimized based on the traditional algorithms, referring to “Chen R, Wang X, Zhang W, et al. A hybrid CNN-LSTM model for typhoon formation forecasting[J]. Geoinformatica, 2019, 23(3): 375-396”.  Besides, there are many studies on the prediction of typhoon types using single and combined neural network models. We summarize the related algorithms and re-carry out experiments under the same conditions to verify the effectiveness of the algorithm proposed in this paper, referring to “Jiang G Q, Xu J, Wei J. A deep learning algorithm of neural network for the parameterization of typhoon‐ocean feedback in typhoon forecast models[J]. Geophysical Research Letters, 2018, 45(8): 3706-3716” and “Hsieh P C, Tong W A, Wang Y C. A hybrid approach of artificial neural network and multiple regression to forecast typhoon rainfall and groundwater-level change[J]. Hydrological Sciences Journal, 2019, 64(14): 1793-1802”. The advantages and disadvantages of the proposed GCN-LSTM algorithm are further clarified by comparing with the latest studies and traditional research algorithms.

Reviewer 2 Report

This is an interesting paper that proposed a new framework of deep learning neural network, Graph Convolution-Long Short-Term Memory Network (GC-LSTM), to predict the typhoon levels based on the Japanese “Himawaril” series of satellite cloud pictures. The experimental result indicates that the proposed framework outperforms the other prediction models via the evaluations of the parameters/indices. Before publishing in the journal of Sensors, there are still minor revisions needed to be addressed.

  1. Lines 327~328, the highest recognition rate is 92.5% when the proportion of neurons is 60%. However, if I don’t misunderstand, I see that Figure 12 shows the highest recognition rate when the proportion of neurons is 70%. Please confirm that.
  2. The contents of the paragraph in lines 349~356 (including Figure 13c) are unclear. For instance, what’s the meaning of actual predictions? Does the actual result mean the reference data? How to see the better prediction stability of the model based on the actual or prediction result ranges? How to evaluate the stability and the error? Please strengthen the explanations about the above concerns.
  3. What are the meanings of the spots appearing In the box plots of Figures 13a and 13b?

Author Response

Point 1: Lines 327~328, the highest recognition rate is 92.5% when the proportion of neurons is 60%. However, if I don’t misunderstand, I see that Figure 12 shows the highest recognition rate when the proportion of neurons is 70%. Please confirm that.

Response 1: We would like to express our sincere thanks for your valuable and constructive comments. Line 327, “60%” is a typo. We are sorry for the carelessness. When the proportion of neurons is 70%, the highest recognition rate is 92.5%.

Point 2: The contents of the paragraph in lines 349~356 (including Figure 13c) are unclear. For instance, what is the meaning of actual predictions? Does the actual result mean the reference data? How to see the better prediction stability of the model based on the actual or prediction result ranges? How to evaluate the stability and the error? Please strengthen the explanations about the above concerns.

Response 2: Thanks for your comment. Line 349, “Act” represents the actual output result on the test dataset, and “Pre” represents the actual output result on the training dataset. We output and analyze all satellite cloud pictures from 2010 to 2019 according to different years. The volatility of the data we mentioned comes from the different output results obtained under the same output conditions. Smaller data fluctuations indicate that the model is more stable and will not produce large deviations in prediction.

Point 3: What are the meanings of the spots appearing In the box plots of Figures 13a and 13b?

Response 3: Thanks for your comment. Spots in Figure 13 (A-B) show the original data distribution. The bars show the statistical results of all the data.

Reviewer 3 Report

The paper proposes a new framework of deep learning neural network, Graph Convolution-Long Short-Term Memory Network (GC-LSTM), which is based on the data of satellite cloud pictures of Himawaril-8 satellite in 2010-2019. To process the irregular spatial structure of satellite cloud pictures the Graph Convolutional Network (GCN) is used.  To learn the characteristics of satellite cloud pictures over time the Long-Short-Term Memory (LSTM) network is employed.

The results presented  show the the model can provide theoretical
basis for the related research of typhoon level classification. Such results are attractive to the research community in this field and therefore I believe that the paper merits publication.

However, a few improvements are necessary before publication, notably:

1) Add a few sentences to the Introduction mentioning the tools of sequential analysis and percolation theory used to study the transition processes in the coupled ocean-atmosphere system. To this aim an instability indicator for the detection of the characteristics of the state for this system, has been proposed (Krapivin et al., 2012). 

Krapivin, V. F., Soldatov, V. Y., Varotsos, C. A., & Cracknell, A. P. (2012). An adaptive information technology for the operative diagnostics of the tropical cyclones; solar–terrestrial coupling mechanisms. Journal of Atmospheric and Solar-Terrestrial Physics89: 83-89.

2) Insert in the Introduction the information to the reader that an information-modeling tracker of tropical cyclones etc based on the cluster algorithm has already been proposed assessing the instability of the ocean-atmosphere system and applied to the cases of Franklin, Harvey, Irma and Katia events (Varotsos et al., 2019; Varotsos and Krapivin 2020).

Varotsos, C. A., Krapivin, V. F., & Soldatov, V. Y. (2019). Monitoring and forecasting of tropical cyclones: A new information-modeling tool to reduce the risk. International Journal of Disaster Risk Reduction36, 101088.

Author Response

Point 1: Add a few sentences to the Introduction mentioning the tools of sequential analysis and percolation theory used to study the transition processes in the coupled ocean-atmosphere system. To this aim an instability indicator for the detection of the characteristics of the state for this system, has been proposed (Krapivin et al., 2012).

Krapivin, V. F., Soldatov, V. Y., Varotsos, C. A., & Cracknell, A. P. (2012). An adaptive information technology for the operative diagnostics of the tropical cyclones; solar–terrestrial coupling mechanisms. Journal of Atmospheric and Solar-Terrestrial Physics, 89: 83-89.

Response 1: Thanks for your comment. We have inserted the above researches in this field in Introduction.

Point 2: Insert in the Introduction the information to the reader that an information-modeling tracker of tropical cyclones etc based on the cluster algorithm has already been proposed assessing the instability of the ocean-atmosphere system and applied to the cases of Franklin, Harvey, Irma and Katia events (Varotsos et al., 2019; Varotsos and Krapivin 2020).

Varotsos, C. A., Krapivin, V. F., & Soldatov, V. Y. (2019). Monitoring and forecasting of tropical cyclones: A new information-modeling tool to reduce the risk. International Journal of Disaster Risk Reduction, 36, 101088.

Response 2: Thanks for your comment. We have inserted the latest research progress in this field in the Introduction, as you have suggested.

Reviewer 4 Report

Interesting work and very interesting results. The paper seems  well written and calibrated, the only two small things that I would like to point out to the authors concern the parameters used in some equations: in eq (1) it is not clear  what "syi" and "si" represent, further, in the following, I would like to highlight the use of the "l" parameter with two different descriptions. Maybe, a table in the end with a summary of all  parameters user could be usefull, in my opinion.

Author Response

Point: The paper seems well written and calibrated, the only two small things that I would like to point out to the authors concern the parameters used in some equations: in eq (1) it is not clear what "syi" and "si" represent, further, in the following, I would like to highlight the use of the "l" parameter with two different descriptions. Maybe, a table in the end with a summary of all parameters user could be usefull, in my opinion.

Response: Thanks for your comment. We have re-described the meaning of the parameters in equations. As for the “l” parameter, the calculation is at the “l” layer, and all the parameters with “l” indicate that they are at the “l” level.

Round 2

Reviewer 1 Report

It is nice to see the authors have addressed most of my review questions. However, there is still a most important part of the proposed method not being explained adequately.

As pointed out in last round of the review, since a main technical contribution of this work is the use of GCN-LSTM, which is opposed to the use of CNN-LSTM as in some existing work, how GCN (in terms of the graph nodes and edges) models and extracts different typhoon features must be explained explicitly. Without this, the work in this paper cannot be understood properly and technically reproduced. The revised paper still did not address this most important issue. The revised description somehow merely just treated GCN as a blackbox for processing typhoon images. The mapping of graph nodes and edges to which relevant typhoon features was not discussed.

Therefore, the paper is still not technically convincing at the moment.

Author Response

We thank all reviewers and editors for the careful reading of our paper, positive affirmations for our work, and the helpful comment.

Response to reviewer1:

We would like to express our sincere thanks for your valuable and constructive comments. We have made further supplement and explanation to your question on lines 96-117, page 3 and lines 221-230, page 6. Please consider our revision version. Thanks!

We have carefully revised the article according to your suggestions. Thanks again for reviewers’ valuable suggestions for this article. Thanks!